# The Evolution of Life Is a Road Paved with the DNA Quadruplet Symmetry and the Supersymmetry Genetic Code

**DOI:** 10.3390/ijms241512029

**Published:** 2023-07-27

**Authors:** Marija Rosandić, Vladimir Paar

**Affiliations:** 1Department of Internal Medicine, University Hospital Centre Zagreb, (Ret.), 10000 Zagreb, Croatia; 2Croatian Academy of Sciences and Arts, 10000 Zagreb, Croatia; paar@hazu.hr; 3Physics Department, Faculty of Science, University of Zagreb, 10000 Zagreb, Croatia

**Keywords:** genetic code symmetry, supersymmetry genetic code table, genetic code evolution, standard genetic code table, DNA quadruplet symmetry, Chargaff’s first parity rule, Chargaff’s second parity rule, asteroid Ryugu

## Abstract

Symmetries have not been completely determined and explained from the discovery of the DNA structure in 1953 and the genetic code in 1961. We show, during 10 years of investigation and research, our discovery of the Supersymmetry Genetic Code table in the form of 2 × 8 codon boxes, quadruplet DNA symmetries, and the classification of trinucleotides/codons, all built with the same physiochemical double mirror symmetry and Watson–Crick pairing. We also show that single-stranded RNA had the complete code of life in the form of the Supersymmetry Genetic Code table simultaneously with instructions of codons’ relationship as to how to develop the DNA molecule on the principle of Watson–Crick pairing. We show that the same symmetries between the genetic code and DNA quadruplet are highly conserved during the whole evolution even between phylogenetically distant organisms. In this way, decreasing disorder and entropy enabled the evolution of living beings up to sophisticated species with cognitive features. Our hypothesis that all twenty amino acids are necessary for the origin of life on the Earth, which entirely changes our view on evolution, confirms the evidence of organic natural amino acids from the extra-terrestrial asteroid Ryugu, which is nearly as old as our solar system.

## 1. Introduction

Ten years ago, our main challenge was to find a solution as to how, in an autonomous system such as the DNA molecule, Chargaff’s second parity rule (CSPR) [1], also called strand symmetry, can function: Why is the relative frequency of some trinucleotides almost identical to the relative frequency of their reverse complement in the same DNA strand of about 100 kb or more? What is common in, for example, ATG (direct) and CAT (reverse complement), or CAA (direct) and TTG (reverse complement)? What has kept such symmetry unchanged for all DNA prokaryotes and eukaryotes during all of evolution? 

The fundamental role of symmetry in a biological system is to decrease disorder (entropy) and to preserve the integrity of the system. Many scientists have investigated the importance of symmetries in biological processes, especially after the discovery of the DNA molecule and the genetic code as basic structures related to life on Earth. The general question is whether symmetries reflect some fundamental “laws” of genome evolution or whether they are a type of statistical pattern [2]. The idea that natural laws are associated with some symmetry is widespread, but the symbiosis of mathematics and natural laws was not fully understood [3,4]. Earlier, in 1918, Emmy Nöther proved her famous theorem by relating symmetry in time to the energy conservation law [5]. As pointed out by Gross [6], Einstein’s great advance was to put the symmetries as a dominant concept in the fundamental laws of physics, to regard the symmetry principle as the primary feature of nature: the symmetry principles dictate the form of the laws of nature. Einstein’s paradigm implies, in general, a broader view on the problem of the evolution of natural laws. For example, the law of energy conservation is a natural consequence of the existence of time symmetry, and not of some kind of evolution. Analogically, according to Einstein’s paradigm, the Supersymmetry Genetic Code (SSyGC), which is unchangeable during whole evolution, could be considered as a natural consequence of physicochemical symmetries with the corresponding mirror symmetry, and not being generated by an evolutionary process.

Jacques Monod attached great significance to symmetry in biology, which must not be understood in purely geometrical connotations, but rather in a much wider sense, identical to that of order within a structure [7]. 

In 1943, Schrödinger proposed in his lecture at Trinity College in Dublin that hereditary material must take the form of an “aperiodic crystal”, implying the presence of symmetries in the structure of DNA. Ever since Nirenberg’s discovery in 1961 (codons code for individual amino acids), scientists searched for symmetries within genetic code. Up to the present discovery of the SSyGC, complete symmetry in the genetic code has not been found, leaving a doubt as to whether the symmetrical nature as the “protector” of order even exists. 

In a framework of symmetry as one of the guiding principles, we discuss a symmetry investigation of the DNA molecule and of genetic code based on programmable biomolecular mediated processes and physicochemical laws, with Einstein’s symmetry paradigm extended to the life sciences. More specifically, here, we discuss how the laws of DNA and genetic code can be considered as being related to physicochemical and mirror symmetries. 

## 2. Classification of Trinucleotides/Codons

Ariadne’s thread on the path of our discovery of DNA and genetic code symmetries was our trinucleotide classification [8,9,10]. Trinucleotides of each DNA genome and codons of the genetic code consist of four nitrogenous bases: two purines (adenine (A) and guanine (G)) and two pyrimidines (cytosine (C) and thymine (T) or uracil (U)). Thus, three of the bases are found in both DNA and the genetic code, whereas thymine is unique to DNA, and uracil is unique to the genetic code. A nucleotide is formed in the cell when the base attaches itself to the 1′ carbon of the sugar and phosphate attaches itself to the 5′ carbon of the same sugar the nucleotide takes its name from.

At first sight, a simple distribution of trinucleotides on A + T rich and C + G rich showed an important basic structural organization of our DNA classification in the form of 20 quadruplets: 10 A + T rich and 10 C + G rich (Figure 1). Each trinucleotide is very important because they represent the basic structure of the DNA molecule, while the genetic code represents the code in the form of codons for 20 natural amino acids, which are the fundamental structure for the protein synthesis of living species. 

From the classification of trinucleotides/codons, a symmetrical relationship between purines and pyrimidines in each quadruplet is the same between direct (D) and reverse (R), as well as between complement (C) and reverse complement (RC). Purines marked as “0” and pyrimidines as “1” for trinucleotides and codons give eight possible combinations from which the first two are in a D↔C relationship, and the other two are in a D↔R relationship: [000 ↔ 111, 010 ↔ 101], [100 ↔ 001, 011 ↔ 110]. The point is that all four members of each quadruplet are specific, and it is free choice which is considered as direct (D). The others are adjusted according to Watson–Crick pairing for complement, reverse, and reverse complement functions (Figure 1). 

In the quadruplet classification of trinucleotides/codons, there arises a dominant role of double mirror symmetry: within each quadruplet as well as between the whole A + T rich and C + G rich groups of quadruplets (Figure 1). This is in accordance with a reviewer’s comment that we received on our earlier work [8] regarding the far-reaching significance of quadruplet classifications. This was the stimulation for our further symmetry investigations.

## 3. Symmetries of DNA Molecule—Chargaff’s First and Second Parity Rules 

In 1951, Chargaff’s first parity rule on nucleotide pairing in the DNA molecule was published [1]. This statement on the equality of frequencies of nucleotides A and T, as well as C and G, in the whole DNA molecule was fully explained by Watson and Crick in 1953 [12], where two chains of DNA are connected by hydrogen bonds: A and T with two hydrogen bonds and C and G with three hydrogen bonds (Watson–Crick pairing).

In 1968, Chargaff´s unexpected second parity rule (CSPR) showed a marked similarity of frequencies also of nucleotides A and T, as well as C and G, within each of the two strands of DNA. It was published as an empirical global rule for long enough DNA segments and not being derived from a compelling principle like Watson–Crick base pairing underlying the first rule [13]. This rule was extended to the similarity of frequencies of oligonucleotides to those of their respective reverse complements within each DNA strand in long enough segments (>100 kb for trinucleotides) [8,9,10,14,15,16,17,18,19,20,21,22,23,24,25,26,27].

Various other names have also been used for CSPR in the literature, such as “strand symmetry”, “intra-strand symmetry”, “word symmetry”, and “inversion symmetry”. According to its meaning, this rule could also be called Chargaff’s nonlocal pairing. For more than 50 years, a conclusive explanation of CSPR was still rather controversial. Namely, up until our discovery, CSPR revealed general species-independent properties and had remarkable implications for some unknown mechanism that seems to be present [18,26].

However, some possible exceptions from CSPR deserved our attention. Firstly, CSPR is not fulfilled in trinucleotide sequences shorter than 100 kb. By further decreasing the sequence length to about 50 kb, the difference between frequencies *f*(D) and *f*(RC(D)) increases, and for smaller lengths, any tendency of *f*(D) and *f*(RC(D)) frequency identity disappears. To solve this problem, we used the empirically estimated minimal length of a genomic sequence (~100 kb) as the reference value for trinucleotides (n = 3). We showed that for the estimate in this minimal sequence length, each trinucleotide must be present ~1500 times. Using this estimate as a gauge, we determined estimates for minimal lengths of oligonucleotides of other orders [10].

Secondly, CSPR gradually disappears when the number of quadruplet oligonucleotides increases. In each human chromosome, the frequencies *f*(D) and *f*(RC(D)) for trinucleotides differ by less than 1%. For higher-order oligonucleotides with up to six constituting nucleotides, this difference gradually increases, and with ten nucleotides, the frequencies *f*(D) and *f*(RC(D)) differ significantly from each other, i.e., CSPR does not hold true anymore. Namely, to prove CSPR for ten nucleotides (n = 10), the minimal length of the DNA sequence must be about 1,600,000,000 bp, and for n = 11, about 6,500,000,000 bp (double the entire human genome) [10]). This shows that CSPR persists for oligonucleotides of up to nine nucleotides in the human genome for its oligonucleotide frequency pattern. Accordingly, the consequence of one of the deviations from CSPR is an insufficient length of investigated DNA sequences depending on the order of mono/oligonucleotides, which creates quadruplets.

Thirdly, our study showed that, for the coding DNA of any human chromosome, CSPR was not satisfied. As the whole human genome has only about 2% of coding DNA, this difference in CSPR does not exceed 1% for the individual chromosome [10].

## 4. DNA Strand Symmetry/Chargaff’s Second Parity Rule (CSPR)

From 64 possible trinucleotides with A, T, C, and G nitrogenous bases, there is one group of 32 trinucleotides that are direct D, and simultaneously the remaining group of 32 trinucleotides comprise their respective reverse complements RC(D). If the frequency of each trinucleotide D from the first group is approximately equal to the frequency of its RC(D) from the second group (difference < 1%), then CSPR is valid for trinucleotides. Strand symmetry reduces the whole DNA genome or a long enough sequence (˃100 kb) to a binary system. Looking at this bidirectionally (5′ → 3′ top strand, 3′ ← 5′ bottom strand), the same combination of D and RC(D) appears in both strands. Therefore, usually only one strand of DNA is analysed, and the term “strand symmetry” is used as a synonym for CSPR (Figure 2A) 

## 5. DNA Quadruplet Symmetry 

The same possible 64 trinucleotides, just as in strand symmetry, are structured in 20 specific quadruplets according to our trinucleotide classification (Figure 1). Each quadruplet always has four members (direct (D), reverse complement (RC), complement (C), reverse (R)), and creates, between both DNA strands, the quartic system with two quadruplet boxes based on Watson–Crick pairing: Qbox_D-RC_ and Qbox_C-R_ (Figure 2B). The quadruplet, as the basic structural symmetry element of DNA molecules, consists of physicochemical purine/pyrimidine symmetry with Watson–Crick pairing and with fascinating mirror symmetry in the Qbox_D-RC_ and Qbox_C-R_ between both DNA strands. At the same time, the quadruplet also consists of mirror symmetry between both quadruplet boxes in each DNA strand (Figure 2B). Thus, each quadruplet consists of structural symmetries, creating an aesthetic form of “butterfly” double mirror symmetry. 

All four trinucleotides in each box have the same frequency, but frequencies between boxes are different (Figure 2C). The exception is symmetric trinucleotides, which have the same frequencies between D and R as well between C and RC. However, frequencies of all four quadruplet members in both strands of DNA for each individual quadruplet are identical regardless of whether, in the case of trinucleotides, they are symmetric or asymmetric (*f*D = *f*RC = *f*C = *f*R). It should be stressed that, regardless of how many times a quadruplet is multiplied, CSPR is not violated and remains integrated in the DNA genome. 

Four bases as mononucleotides (4^1^) have only two quadruplets: one composed of A and T and the other of C and G nucleotides. Dinucleotides with sixteen combinations (4^2^) have only six different quadruplets. Both have restricted information content. On the other hand, oligonucleotides composed of four nucleotides and 256 combinations (4^4^) give 68 possible quadruplets, which are complicated for this analysis [10]. In this sense, the genome is being gauged for 64 possible trinucleotides (4^3^), which, in the genetic code, represent 61 codons and three stop signals (UGA, UAG, UAA). Trinucleotides have two matrices with 10 A + T rich and 10 C + G rich quadruplets as our quadruplet classification of trinucleotides/codons (Figure 3).

We show that the logarithmic relationship between the oligonucleotide order and minimal DNA sequence length (about 100,000 bp) to establish the validity of CSPR automatically follows from the quadruplet structure of the genomic sequence (Figure 3) [10]. Performing our quadruplet frequency analysis of all complete human chromosomes, for a random 200,000 bp sequence of each chromosome, and for the Neuroblastoma Break Point Family (NBPF) genes that code for Olduvai protein domains in the human genome [28], we show that the coding part of DNA (less than 2% of the whole genome, ~17,000,000 bp in chromosome 1) violates the CSPR. Opposite of that, the 98% non-coding part and the whole human genome agree with CSPR as well as with DNA quadruplet symmetry.

## 6. The Natural Law of DNA Creation and Conservation 

After the discovery of DNA quadruplet symmetry, a more complex problem led us to investigate the genome itself, i.e., how is it possible that each DNA species, despite mutations during evolution, preserves genome symmetries in the form of CSPR and quadruplet symmetry? This question was a large challenge for scientists from the very beginning of CSPR discovery [13]. Sueoka (1995) [29] and Lobry (1995) [30] tried to answer this question independently. Namely, if the strand equivalence holds as Chargaff’s first parity rule and the substitutional dynamics have sufficient time to reach their equilibrium, then CSPR also becomes valid. Forsdyke and Bell (2004) [31] suggested that CSPR reflects the evolution of genome-wide stem-loop potential. Albrecht-Buehler (2007) [18] developed the hypothesis that inversions and inverted transpositions could be a major contributing, if not dominant factor, in the validity of CSPR. 

We discovered the concept of the natural law of DNA creation and conservation owing to our fundamental DNA quadruplet mirror symmetry, which automatically leads to CSPR. The natural law is activated, according to which the same mono- or oligonucleotide insertion must be inserted simultaneously into both strands of DNA. However, regardless of the localization in the second strand, the new DNA segment with stable quadruplet symmetries in a bidirectional 5′3′↔3′5′ manner is created (Figure 4) [9,10]. In this way, identical complementary base pairs are inserted simultaneously in DNA strands, creating quadruplets with strict purine–pyrimidine symmetry, direct–complement symmetry on the principle of Watson–Crick pairing, and underlying mirror symmetry. Consequently, only mutations on the principle of the natural law of DNA creation and conservation could have been incorporated into the genome during evolution without violating the symmetries in creating new species. Those mutations that enter the genome accidentally without obeying the natural law usually are liding to pathological processes and endanger the existence of individual species. For example, single-strand RNA viruses such as coronaviruses are not protected with quadruplet symmetry and are subjected to frequent mutations with the creation of new sorts.

## 7. The Supersymmetry Genetic Code Table

Ariadne’s thread of our investigation leads us to the fundamental discovery of physicochemical symmetries of the genetic code and, in this way, closing *circulus vitae*. From Nirenberg’s discovery in 1961 [32] in which codons code individual amino acids, it has been a 60-year challenge for biologists to find the optimal symmetry of the genetic code. The Standard Genetic Code (SGC) [33] and all other known genetic code tables structured on the U-C-A-G principle suffer from an inability to show the complete physicochemical symmetry between codons. The SGC table has only alphabetic symmetry between all bases (A, G purines; U, C pyrimidines) and is only an aesthetic category [11,34]. The SGC table consists of 4 × 4 boxes with four codons in each box. The problem is that the third base in all sixteen boxes is in a U-C-A-G manner. Therefore, the role of the third base of codons was ignored in the search of symmetries and each box was differentiated only according to the first two bases of each codon. Furthermore, sextets Arginine and Serine have scattered codons.

The stereochemical theory postulates that the structure of the code is determined by a physicochemical affinity between amino acids and codons or anticodons [35]. Unfortunately, symmetries within the genetic code with a different distribution between codons and amino acids in a circular, triangular, rectangular, or torus form and in the binary transformation of nucleotides within codons all suffer from an inability to illustrate the functional physicochemical relationship between codons and amino acids [36,37,38,39,40,41,42,43,44,45]. With respect to the polar requirement, marked differences were observed for the hydrophobicity and lipophilicity parameters encoded by the codon second base of the SGC table [44]. Unfortunately, the result showed only a partial solution related to the physicochemical properties between codons and amino acids. The study of all codes of life with standard methods of science is a new field of research that must be turned into practice (Barbieri, 2014) [46]. Namely, there are about 10^84^ possible codon combinations of the genetic code [36]. 

The genetic code is degenerate because more than one type of codon (2, 3, 4, or 6) may encode a single amino acid. The intriguing algebraic approaches to the genetic code evolution through the progressive symmetry breaking theory explained the observed degeneracy of the genetic code with a mathematical technique for organizing the group theoretical structure [47,48,49,50,51,52,53,54,55]. The evolution of the genetic code through progressive symmetry breaking proposes that, in the beginning, it was not possible to distinguish the function of codons, which therefore all encode the same information. With the consecutive creation of amino acids during such a proposed evolution, the symmetries among codons, i.e., the number of codons within degeneracy groups, gradually decrease (two singlets, nine doublets, two triplets, five quadruplets, and three sextets). Because of this, the symmetric pattern of codon degeneracy is supported with a unified mathematical framework by using the group theoretical structure [56,57]. In conclusion, such a degeneracy distribution through the progressive symmetry breaking theory takes into consideration only one input—the number of codons for each amino acid—according to Nirenberg’s empirical result, and without any physicochemical affinity between codons and amino acids. 

The evolution of the genetic code and life on Earth was a scientific challenge for many scientists, with interesting results, but without discovering complete physicochemical genetic code symmetries [2,33,35,44,46,48,49,58,59,60,61,62,63].

A completely new approach is our discovery of the fascinating supersymmetry genetic code (SSyGC) table with five physicochemical symmetries between bases, codons, and amino acids: (1) purine–pyrimidine symmetry on the principle of Watson–Crick pairing (A↔U, C↔G), (2) direct–complement symmetry between codons, (3) double mirror symmetry between bases and codons, (4) A + T rich and C + G rich symmetry between codons, and (5) symmetry between the position of amino acids (Figure 5). 

During our discovery of the SSyGC table, we conformed to the traditional concept with 4 × 4 boxes as in the SGC table [33]. In this way, we created the ideal symmetry genetic code table [34], which has all four columns as the standard genetic code table with the same distribution of purines and pyrimidines, but Leucine had scattered codons. Only a form of 2 × 8 boxes of the SSyGC table revealed superior and dominant double mirror symmetry, which we named “supersymmetry”. Also, in the SSyGC table, for the first time between many genetic codes, all three sextets have codons in continuity and are completely mutually connected (Figure 5). Therefore, they showed a path for all other symmetries in the SSyGC table.

Namely, our path to the ordering of codons started from Serine because we noticed the unique regularity: its six codons are in two neighbouring boxes that are in a direct–complement relation. In this way, we discovered the logic of the third base in codons and the link between Serine and Arginine in the same column of boxes and between Serine and Leucine in the neighbouring column of boxes. After identifying this core structure of the SSyGC table, the further ordering of other codons according to a similar logic is straightforward.

It is fascinating that the SSyGC table starts with a very important AUG start signal. This is an automatic result of the SSyGC symmetries. Namely, the position of each codon is unique and strictly localized. At the same time, all three stop signals (UAG, UAA, UGA) are positioned according to the mirror symmetry in both halves of the code in two boxes with the same 1-0-0 purine–pyrimidine relationship (Figure 5).

Our SSyGC table also consists of 16 codon boxes, as in SGC, but in the form of two columns, each with eight boxes, and with the same codons in each box like the SGC but in a different arranging. The main difference in the SSyGC table is the position of the third base between direct and complement codon boxes on the principle of Watson–Crick pairing: GACU ↔ CUGA, AGUC ↔ UCAG. This leads to the physicochemical code symmetry with the double mirror symmetry (Figure 5).

We postulate “the symmetry theory of genetic code”, which is based on the unique physicochemical purine–pyrimidine symmetry net between bases and codons of our SSyGC table. Because of the unique symmetry net, the position of each codon in the SSyGC table is strictly definite. The common purine–pyrimidine symmetry net as “the golden rule” of the SSyGC table, which is identical and universal for all RNA and DNA species on Earth, has remained unchanged during all of evolution. It is also identical for more than 30 known slightly alternative nuclear and mitochondrial genetic codes, including those that will be discovered in the future. We show that variations of the number of codons for individual amino acids inserted in the SSyGC table arise most often with a capture from the neighbouring codons from the split box according to the metabolic requirements [11,34]. This is valid also for the mitochondrial invertebrate code (Figure 6), the mitochondrial trematode code, the mitochondrial echinoderm and flatworm code, and the mitochondrial alternative flatworm code. Among other differences with respect to the standard genetic code, they even have eight codons for Serine, which is the largest number of codons among all amino acids. Namely, Serine is a building amino acid in most proteins and participates in the regulation of energy metabolism and fuel storage in the species body. The life cycle of these species demands additional energy that is enabled by Serine.

The symmetry net structure of the SSyGC table enables an automatic transformation with the direct alignment of all codons from direct boxes (top strand) and all codons from complement boxes (bottom strand) of the genetic code into a DNA-type sequence with Watson–Crick pairing, as well as its form analogous to the 5′3′ codon and 3′5′ anticodon: 

Codons from the first column of the SSyGC table:



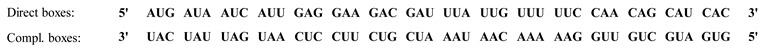



Codons from the second column of the SSyGC table:



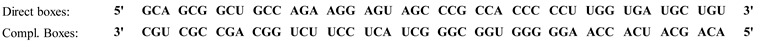



The perfect symmetry of each individual codon is clearly visible. For example, all symmetric codons such as UUU-AAA, CCC-GGG, CAC-GUG, UGU-ACA, etc., strictly aligned one below the other in all four rows because of mirror symmetry. The horizontal third base alignment from direct boxes is GACU, GACU, AGUC, AGUC, AGUC, AGUC, GACU, and GACU with their complement in complement boxes. Therefore, it follows that the SSyGC table and DNA molecule are like a coin with two sides.

It seems that codon–anticodon symmetry as a complete structural physicochemical form of the unique SSyGC table made possible, in the early Earth, the synthesis of an immature protein with anticodon stem-loop tRNA, later becoming double-stranded DNA; this is also the theoretical approach from earlier investigations [62,63].

RNA viruses have an identical SSyGC table with the unique common purine–pyrimidine symmetry net, like DNA species. This means that the symmetry net had a direction for RNA to DNA transformation on the principle of Watson–Crick pairing, which enabled the fingerpost for evolution from an RNA to DNA molecule. 

Translation is not limited to twenty amino acids. The additional Selenocysteine as the 21st amino acid takes possession of the UGA stop signal and Pyrolysine as the 22nd amino acid [64] takes the UAG stop signal, but the SSyGC table and the basic purine–pyrimidine symmetry net remains unchanged.

One could assume that one new non-natural amino acid, because of a special metabolic need of species (or experimentally), adopts the codon of some other natural amino acid or stop signal of code without disturbing the physicochemical symmetry net of the SSyGC table. However, there is a question of the consequences of such artificial translation, with possible harmful or even lethal consequences.

Fredens and coworkers (2019) [65] constructed laboratory *Escherichia coli* with the entire synthetic DNA genome that utilized just 61 codons for protein synthesis compared to 64 in natural living organisms. Performing “synonymous codon compression”, they recompiled the *E. coli* genome with two (TCG, TCA) out of six codons encoding Serine and the TAG stop signal. Accordingly, this synthetic organism with “laboratory different genetic code” uses 59 codons and TAA and TGA to encode 20 amino acids and two stop signals. *E. coli* with “synthetic DNA” displayed only minor changes, with a slower growth rate, slightly elongated cells, and enabled deletion of previously essential tRNA. If the two codons and one stop signal are omitted from the SSyGC table, the remaining codons and stop signals stay in the same positions within the common purine–pyrimidine symmetry net, keeping their symmetrical mutual relationship. The positions of omitted codons and stop signals remain empty, because none of the remaining codons and stop signals can replace them without violating the symmetries, as each codon and stop signal has a strictly defined position within the SSyGC table. In this way, it is not a different genetic code but unfortunately an abnormal, mutilated, and artificial SSyGC table [11]. Consequently, the synthetic *E. coli* could survive, but with mutation marks regarding the shape and growth rate. *E. coli* cannot survive without all codons of any amino acids. 

An important example is the protist *Xerella Beyerinck,* a single-cell green microalgae and one of the oldest living species with an estimated age of about 2.5 billion years. It also has the same SSyGC table, which consists of 50% proteins and possesses all twenty natural amino acids [11,66]. The smallest RNA virus identified to date is the human hepatitis D virus—its genome being only 1.7 kb in size—and it also has the same SSyGC table with the purine–pyrimidine identical symmetry net. Namely, each small RNA or DNA genome that has all 64 possible combinations of trinucleotides possess the complete code of life in the form of the SSyGC table and the purine–pyrimidine symmetry net with codons for all twenty natural amino acids. 

## 8. During Sixty Years, What Has Hindered the Discovery of Complete Physicochemical Symmetries of the Genetic Code?

In 1961, Nirenberg and collaborators deciphered the genetic code, determining experimentally which codons correspond to each of the 20 natural amino acids, but without considering the codon’s regularity in the form of a genetic code table. This challenge was addressed by Nobel laureate F. H. Crick, and in 1968, a solution was published [33] under the name Universal Genetic Code table and readily included in biology and genetics textbooks. In search of symmetry in this genetic code table, the guideline was Watson–Crick A↔T and C↔G base pairing, which was discovered in 1953 for the structure of the DNA molecule. This goal was achieved as a half-way result only, and the whole concept of the creation of the genetic code table was comparison completed with Crick’s random “frozen accident hypothesis”.

In the meantime, more than thirty different codes have been discovered for genomes of some bacteria and archaea as well as for some organellar mitochondrial and eukaryotic nuclear genomes. Because of this, the name of the Universal Genetic Code was changed to the Standard Genetic Code.

The fact that the Standard Genetic Code is degenerate, i.e., that more than one codon can code for the same amino acid, the search for symmetries cannot be completely successful with amino acid arrangement. We realized that the key to the genetic code symmetries must be between codon purines and pyrimidines as the starting point. This approach led us to the discovery of the supersymmetry genetic code (SSyGC) table, characterized by codon physicochemical symmetries, also including the double mirror symmetry. The symmetry core of the SSyGC table is the purine–pyrimidine symmetry net as a “golden rule”, which is common for all RNA and DNA species and unchangeable during evolution.

The Standard Genetic Code has an ordering of codons according to pyrimidines (U, C) and purines (A, G). This ordering is strictly established both in horizontal and vertical directions. Thus, the resulting genetic code has only alphabetic symmetry, with a UCAG ordering of bases. Therefore, the third base in codons does not differentiate among sixteen codon boxes in the Standard Genetic Code table and cannot fully contribute to the discovery of physicochemical symmetries of genetic code. This has, to this day, contributed to the problem of many studies based on the Standard Genetic Code table with a UCAG ordering of the third base [55,60,61,67,68,69,70,71,72,73,74,75,76,77,78,79,80,81,82,83,84,85,86,87,88,89,90,91,92,93,94,95,96,97,98,99,100,101,102]. For example, the hydrophobicity and lipophilicity of amino acids have some symmetry relationship only with the second base of codons [44]. The result was also studied using algebraic approaches to the degeneracy of genetic code and hypothesis of the evolution of genetic code through progressive symmetry breaking [47,48,49,50,51,52,53,54,56]. 

On the contrary, with our SSyGC table, we have proved that the third base is a crucial point for the discovery of physicochemical symmetries on the principle of Watson–Crick base pairing (A↔U and C↔G like codon/anticodon) and for the discovery of double mirror symmetry. 

We have also proved that the DNA quadruplets have the same symmetries, as well as our classification of trinucleotides/codons (Figure 1). It is very important and unique for our SSyGC table that the symmetries of the SSyGC table are organized on the principle of direct–complement (codon–anticodon); this enables the direct transformation of the SSyGC table into the DNA molecule with double mirror symmetry as well. 

Because of the complete physicochemical symmetries of the SSyGC table, it is not necessary to involve “the frozen accident”. All more than thirty alternative genetic codes with a slight departure from the standard code can be incorporated in the SSyGC table (Figure 6). Thus, the SSyGC table fulfils all physicochemical criteria on the origin of the genetic code (2, 34, 62). Of special importance is the meaning of complete physicochemical symmetries in the SSyGC table, common for all living RNA and DNA species and unchanged during the whole evolution. In this theoretical approach, there is no evolution of the genetic code, but instead it has a power of the natural law in an analogy to Nöther՚s theorem [5] for the natural law of energy conservation. 

Our symmetry-based theory of genetic code broadens the horizon for understanding evolution as a fundamental process in creation and the richness of life on Earth. This approach points out that besides mutation and natural selection, other factors also may have been responsible for evolution, as is the basic and unchangeable physicochemical purine–pyrimidine symmetry net as “the golden rule” of the genetic code structure.

We point out that the fundamental role of symmetry in the genetic code is to decrease disorder (information entropy) and to preserve the integrity of a biological system [8,9,10,11,27,34] during evolution in a way of extending Einstein’s paradigm, putting symmetries as a dominant concept in the fundamental law of physics, to the phenomenon of life. One impressive case of the realization of Einstein’s paradigm in physics is the famous Nöther theorems: Emy Nöther proved mathematically that the law of energy conservation is a consequence of the time symmetry [5].

## 9. Discussion

Regarding the results of our discovery of the multifaceted physicochemical symmetry of the SSyGC table remaining unchanged during all of evolution, we hypothesize that life on Earth was developed when all natural amino acids were already present [11]. We conclude “that the fascinating physicochemical unchangeable symmetry net as ‘the golden rule’ of symmetry of the genetic code table is an argument against the random gradual and individual development of amino acids during the early evolution in the creation of life”. Our manuscript was published online on 14 May 2022 [11]. Our hypothesis received scientific support less than 1 month later, on 9 June 2022, when the fascinating results were published from 16 costly representative particles brought directly to Earth by the Hayabusa2 spacecraft from asteroid Ryugu, which is aged about 4.6 billion years like our solar system [103]. Sample return missions with twelve prebiotic amino acids between 9% organic materials represent great opportunities to study materials from known locations on the targeted extra-terrestrial body [104] without uncontrolled exposure to the atmosphere of Earth and biosphere and to change our view about the creation of life on Earth. 

An especially important discovery on the asteroid is also the presence of L-2-aminobutyric acid as a chiral precursor for the synthesis of non-natural L-amino acids in the later stage of origin [62,63,105]. The discovery of extra-terrestrial abiotic amino acids confirms our hypothesis that for the origin of life on Earth, all 20 natural amino acids were necessary. There are scientific arguments that they were already present at the time of the creation of the solar system [106]. 

Therefore, the first manifestation of life in the form of a single-stranded RNA molecule already had genomes with the complete code of life in the form of the SSyGC table for proteinogenesis, as well as instructions as to how to develop the DNA molecule on the principle of Watson–Crick pairing. Similarly, we discovered that DNA and genetic code with unchangeable symmetries during evolution decrease the disorder (entropy) of vital processes and preserve the genome’s integrity, which changes our view on all of evolution. After the discovery of natural amino acids on the asteroid Ryugu, the connecting of the genetic code degeneracy with symmetries seems to lose sense. Namely, if the prebiotic organic material was originating during the time of the creation of the solar system, then the natural amino acids were delivered to the early Earth at the time of the origin of life. Then, one would expect that each amino acid was coded by a different number of codons in accordance with the metabolic need of species; this is reflected in the structure of the genetic code.

At the end of this journey investigating the symmetry of the genetic code and DNA molecule, we can ask: what would be the answer of biology, molecular biology, genetics, or medicine students to the question of what could be said about the Standard Genetic Code table? The answer could be that it is a table that consists of 4 × 4 boxes containing 61 codons and three stop signals ordered alphabetically as U-C-A-G (two pyrimidines and two purines), both horizontally and vertically. Simultaneously, besides the Standard Genetic Code table, there are also more than 30 alternative nuclear and mitochondrial genetic codes. There are many aspects of SGC in the recent literature, but without complete physicochemical symmetries between codons [33,35,36,55,58,59,60,61,67,68,69,70,71,72,73,74,75,76,77,78,79,80,81,82,83,84,85,86,87,88,89,90,91,92,93,94,95,96,97,98,99,100,101,102,107]. 

On the other hand, answers to the question regarding the characteristics of the SSyGC table would be completely different:The SSyGC table starts with an AUG start signal.All 61 codons and three stop signals are arranged in 2 × 8 boxes, which alternate in each row on the principle of A + T rich and C + G rich codons but with the same purine and pyrimidine ordering.The ordering of purines and pyrimidines in both columns is identical.Vertically, between direct and complement boxes, purines, and pyrimidines as well as codons are regularly ordered on the principle of Watson–Crick pairing.Horizontally in the same row, purine in the first column transforms in the purine of the second column, and analogously pyrimidine transforms in pyrimidine, creating alternate A + T rich and C + G rich codons.In the SSyGC table, the codons of all amino acids are not scattered, including three sextets for Serine, Arginine, and Leucine.The SSyGC table has a central vertical and horizontal double mirror symmetry according to the mirror symmetry axis as well as a double mirror symmetry of DNA quadruplets and a classification of trinucleotides/codons.In this way, the physicochemical unique symmetry net of the whole SSyGC table is structured, creating full symmetries between bases, codons, and amino acids.The symmetry net is unique and common for all RNA and DNA living species on Earth.The symmetry net is also common for more than 30 nuclear and mitochondrial genetic codes, which differ from the Standard Genetic Code table.Due to the symmetry net, codons of the SSyGC table directly transform in the DNA molecule with Watson–Crick pairing (32 codons and 32 anticodons, direct and their complement, respectively).The unique symmetry net has remained unchanged during evolution and has the power of the natural law for the origin of life.The protection of the genetic code and DNA molecule symmetries during all of evolution reveals their role in decreasing entropy (disorder) and the preservation of species integrity.

Now, 60 years after Nirenberg’s empirical discovery, physicochemical symmetries of the SSyGC table for all RNA and DNA species on Earth are discovered, shedding, together with DNA symmetry, a new light on the evolutionary development of species.

The recent discovery of extra-terrestrial abiotic amino acids with identical physicochemical symmetries to the genetic code and DNA molecule, and at the same time, the discovery of unchangeable genetic code during evolution for all RNA and DNA species are a challenge for scientists of genetics, biology, chemistry, physics, medicine, and philosophy to investigate and reveal the riddle of the origin of life.

## Figures and Tables

**Figure 1 ijms-24-12029-f001:**
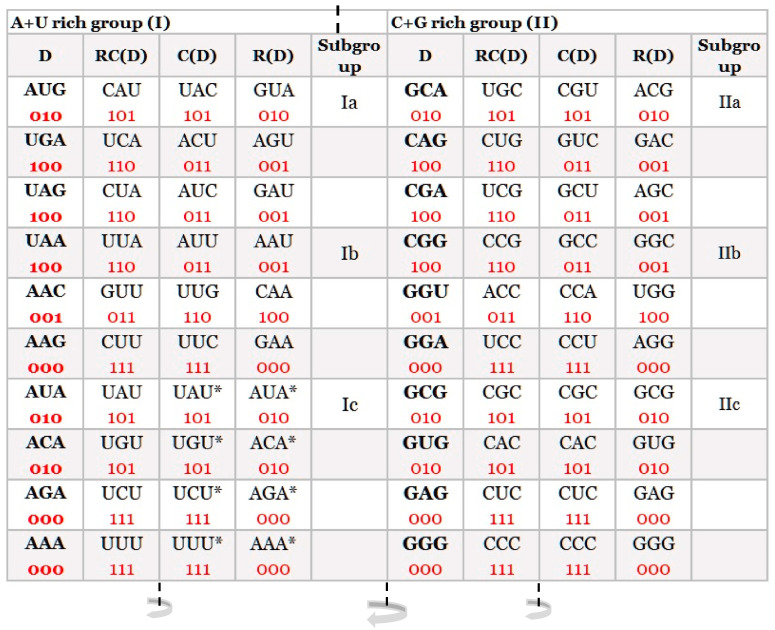
Our quadruplet classification of 64 codons (with U—uracil) for the genetic code, or trinucleotides (with T—thymine instead of uracil) for RNA and DNA genomes. Each quadruplet is unique and consists of four specific codons or trinucleotides denoted as direct D, reverse complement from direct RC(D), complement from direct C(D), and reverse from direct R(D). Ten A + U rich (group I) and ten C + G rich (group II) quadruplets are organized in three subgroups. Ia consisting of nonsymmetrical codons/trinucleotides containing three different nucleotides, Ib consisting of nonsymmetrical codons/trinucleotides containing two different nucleotides, and Ic consisting of symmetrical codons/trinucleotides that contain duplicated codons/trinucleotides labelled with an asterisk (D = RC, C = R). The first four A + U rich quadruplets were generated with start/stop signals: AUG, UGA, UAG, and UAA. The C + G rich trinucleotides correspond to the purine–purine and pyrimidine–pyrimidine transformation of A + U rich codons/trinucleotides. Three symmetries are present in our codon/trinucleotide classification: (1) purine–pyrimidine symmetries in each quadruplet, (2) purine–pyrimidine symmetries within and between A + U rich and C + G rich quadruplets in the same row of the classification, and (3) mirror symmetry between the direct-reverse and complement-reverse complement in the same quadruplet. Mirror symmetry is also present between purines and pyrimidines of the whole A + T rich group and C + G rich group of codons/trinucleotides. For clarity, the white and grey rows are alternating, to emphasize pairs of A + T rich and C + G rich codons. 0, purine; 1, pyrimidine. It is irrelevant which codon/trinucleotide in the quadruplet is direct, because the other three are accordingly adapted: mirror symmetry. From work by Marija Rosandić and Vladimir Paar [11], published by Elsevier and reproduced with the permission of the publisher.

**Figure 2 ijms-24-12029-f002:**
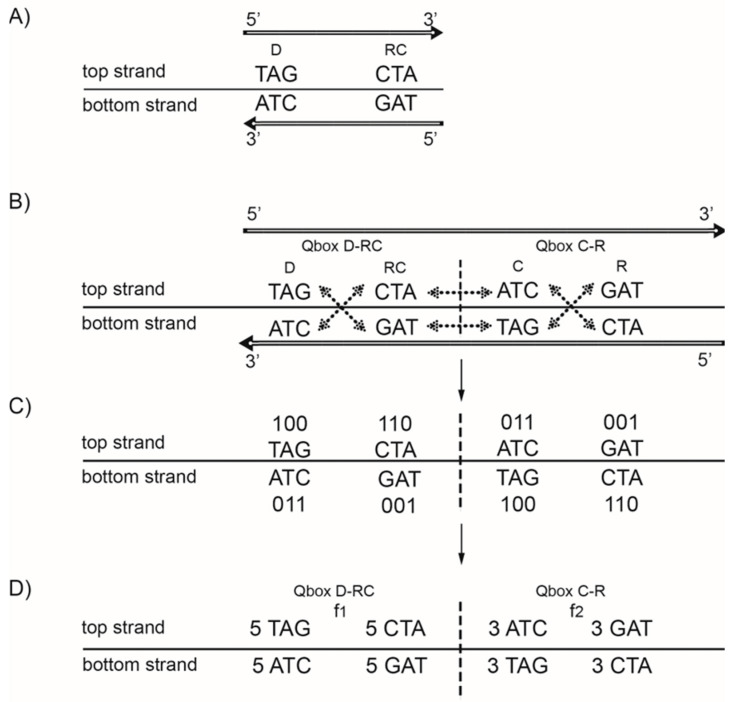
The difference between strand symmetry (CSPR) and quadruplet symmetry for triplets. (**A**) Strand symmetry includes the same strand direct (D) and reverse complement (RC) of a triplet. Reading bidirectionally, in the direction of the arrow, the same trinucleotides appear in both strands and DNA is reduced to a binary system. However, in this way, quadruplet symmetries among trinucleotides are not evident. (**B**) Quadruplet symmetry includes all four members of the whole quadruplet of trinucleotides: direct (D) and reverse complement (RC) as well as complement (**C**) and reverse (R) in both strands of DNA as a quartic system. The quadruplet boxes Qbox _D-RC_ and Qbox _C-R.R_ have their own mirror symmetry between both strands of DNA. The mirror symmetry is present also between both Qboxes in each strand. Thus, each quadruplet consists of structural physicochemical symmetries, creating an aesthetic form of “butterfly” double mirror symmetry. (**C**) The same quadruplet mirror symmetries are present in the purine–pyrimidine relationship: 0 is assigned to purines (A, G), and 1 is assigned to pyrimidines (T, C). (**D**) All four members of the same Qbox have the same frequencies (*f*D = *f*RC, respectively, *f*C = *f*R), but frequencies between the Qboxes differ mutually. For quadruplets with symmetric trinucleotides, such as AGA or CTC, there is no difference in frequencies between boxes. However, frequencies in both strands of DNA for each individual member of the quadruplet are identical regardless of whether the trinucleotides are symmetric or asymmetric as well as whether the four members of each quadruplet are a mononucleotide, dinucleotide, trinucleotide, or oligonucleotide (*f*D = *f*RC = *f*C = *f*R). From work by Marija Rosandić and Vladimir Paar [10], published by MDPI.

**Figure 3 ijms-24-12029-f003:**
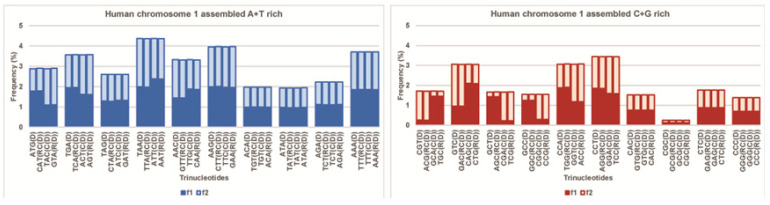
A + T rich and C + G rich trinucleotide quadruplet matrices with relative frequencies of trinucleotides from human chromosome 1. In each quadruplet, the frequency of all four members in both strands is identical (*f*D = *f*RC = *f*C = *f*R), noted as a plateau on the upper edge of each quadruplet. The plateau shows that the investigated sequence (chromosome, genome) is in accordance with Chargaff’s second parity rule. From work by Marija Rosandić and Vladimir Paar [10], published by MDPI.

**Figure 4 ijms-24-12029-f004:**
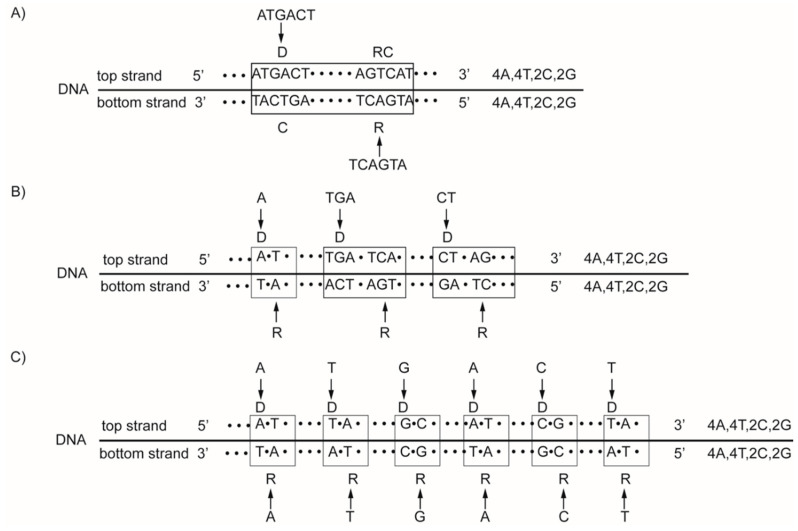
Examples for the natural law of DNA creation and conservation. According to this law, all mono/oligonucleotides that enter one strand of DNA must enter the second strand regardless of their localization. Binding with a complementary pair, the quadruplet structures with mirror symmetry between both strands and the final CSPR are created. At the same time, the new DNA segment in a bidirectional 5′3′↔3′5′ manner is also created. The total number of bases in both strands is identical. (**A**) Example with the entrance of 6 oligonucleotide ATGACT into the top strand, its reverse oligonucleotide TCAGTA entering the bottom strand. (**B**) The same nucleotides may also enter as mononucleotide (A), trinucleotide (TGA), and dinucleotide (CT). The farther process and result is identical, as in (**A**). (**C**) The same 6 nucleotides can enter the top strand and the bottom strand individually as mononucleotides. Binding with a complementary pair, the quadruplet structures with mirror symmetry are created, and the final CSPR result is identical as in A and B. From work by Marija Rosandić and Vladimir Paar [10], published by MDPI.

**Figure 5 ijms-24-12029-f005:**
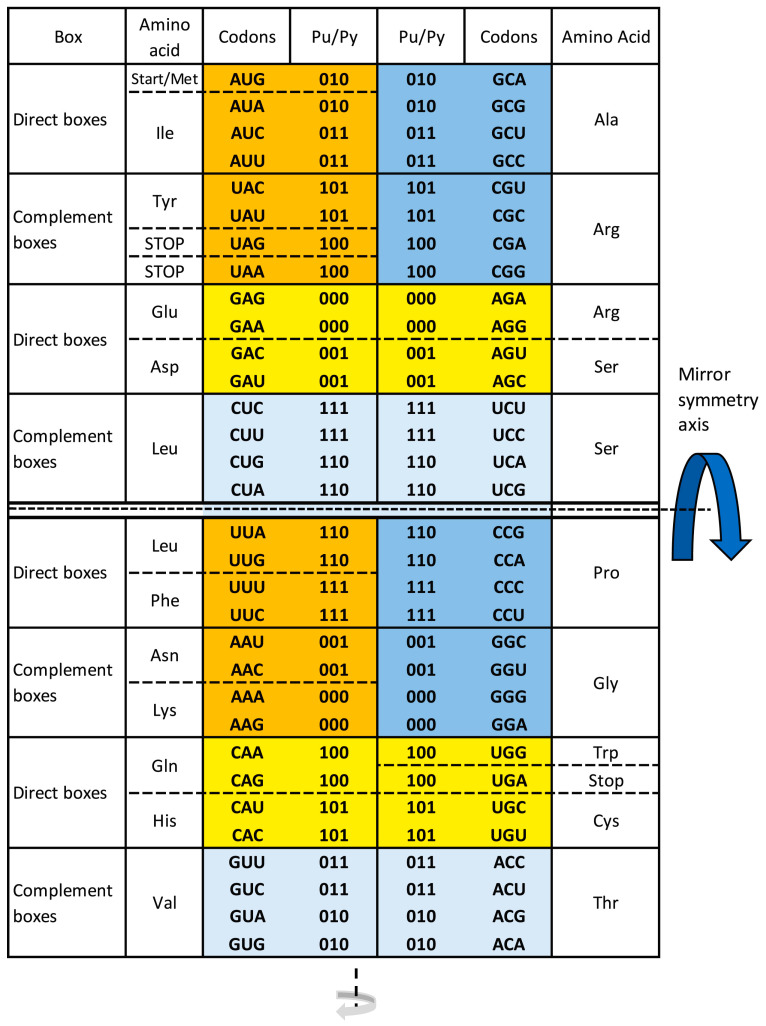
The supersymmetry genetic code (SSyGC) table with 2 × 8 boxes. The AUG start signal is at the beginning of the SSyGC table. It has the same distribution of the purine/pyrimidine profile in both columns, and simultaneously the same profile distribution pairs of codon rows within each box. There are five symmetries present: purine–pyrimidine symmetry between bases and codons, direct–complement symmetry of codons between boxes, and A + U rich and C + G rich symmetry of codons between two columns. Superior dominant double mirror symmetry as a core symmetry of the SSyGC table is present between all purines and pyrimidines of the whole genetic code. With the horizontal and vertical central mirror symmetry axis, it created the purine–pyrimidine symmetry net as “the golden rule “for all RNA and DNA species that is unchangeable during evolution. Purines A and G are marked as 0, and pyrimidines C and U are marked as 1. Mirror symmetry with the horizontal symmetry axis is also present between the second and third bases of codons. Mirror symmetry simultaneously generated symmetry between positions of amino acids. In such a way, the sextets for Serine, Arginine, and Leucine, each with six codons, are, for the first time, positioned in continuity—0 pu, purine; 1 py, pyrimidine; bold black line, the axis of the mirror symmetry; dark yellow, two pairs of split boxes with direct–complement symmetry between codons; dark blue, two pairs of no-split boxes with direct–complement symmetry between codons; light yellow, two pairs of split boxes with purine ↔ purine, pyrimidine ↔ pyrimidine transformation between codons of both columns; and light blue, two pairs of non-split boxes with purine ↔ purine, pyrimidine ↔ pyrimidine transformation between codons of both columns as in the whole code.. From work by Marija Rosandić and Vladimir Paar [11] published by Elsevier and reproduced with the permission of the publisher.

**Figure 6 ijms-24-12029-f006:**
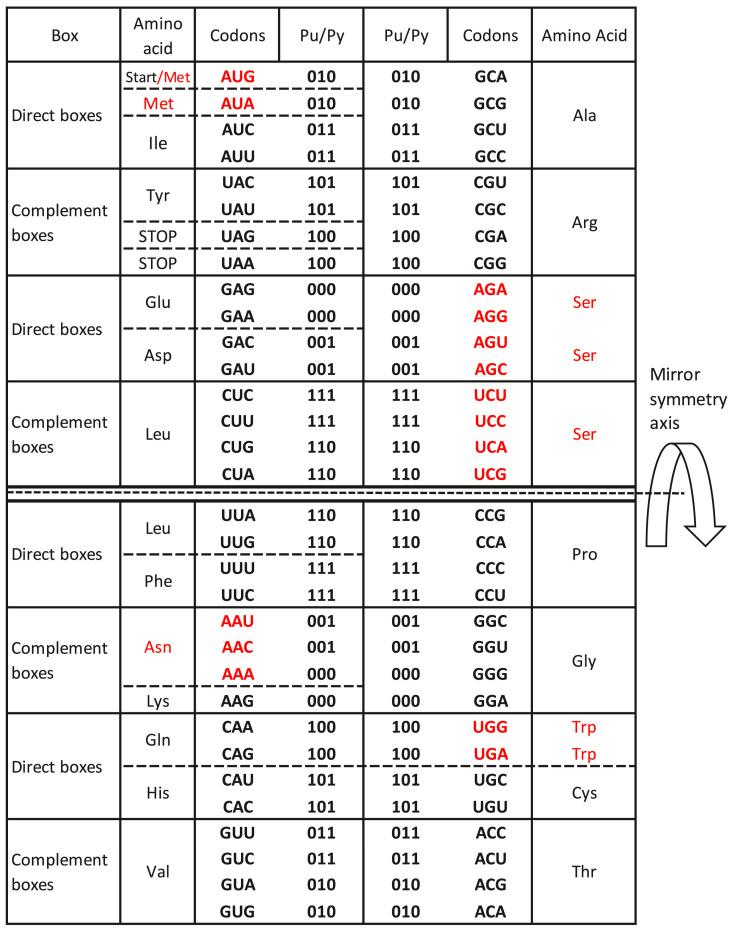
The mitochondrial invertebrate code incorporated in the SSyGC table. Red: Methionine (M, Met) expands to the neighbouring Isoleucine (Ile) codon AUA; Arginine (Arg) AGA and AGG codons become the 7th and 8th codons for Serine (Ser). In various nuclear and mitochondrial genetic codes, individual amino acids usually capture a codon from a neighbouring amino acid in the SSyGC table [11]. But the purine–pyrimidine symmetry net always remains unchanged. From work by Marija Rosandić and Vladimir Paar [11], published by Elsevier and reproduced with the permission of the publisher.

## Data Availability

The data are contained within the article.

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
