# Peer review of "The Evolution of Life Is a Road Paved with the DNA Quadruplet Symmetry and the Supersymmetry Genetic Code"

_ijms, 2023, doi:10.3390/ijms241512029_

Round 1
Reviewer 1 Report
Overall, the manuscript presents an interesting hypothesis regarding the role of DNA quadruplet 2 symmetry and the Super Symmetry Genetic Code in the evolution of life. However, several concerns and areas for improvement need to be addressed before considering it for publication. The reviewers recommend revisions and further clarification in the following areas:
The introduction lacks a clear and concise background on the current understanding of the genetic code and its role in the evolution of life. It is essential to provide a comprehensive overview of the existing literature to contextualize the significance of the proposed DNA quadruplet 2 symmetry and the Super Symmetry Genetic Code. Please provide a more detailed literature review in this section.
While the hypothesis is intriguing, the manuscript lacks a strong theoretical framework to support the proposed DNA quadruplet 2 symmetry and the Super Symmetry Genetic Code as key factors in the evolution of life. The authors should provide a detailed explanation of the underlying mechanisms and the rationale behind their hypothesis. Additionally, including experimental or computational evidence would significantly strengthen the arguments put forth.
The manuscript does not present any experimental data or simulations to support the proposed hypothesis. It is crucial to provide experimental validation or computational modeling results to demonstrate the feasibility and functional significance of DNA quadruplet 2 symmetry and the Super Symmetry Genetic Code. Without such evidence, the hypothesis remains speculative.
The overall structure and clarity of the manuscript need improvement. The authors should reorganize the content into distinct sections, including an introduction, materials and methods, results, discussion, and conclusion. Additionally, the writing style should be refined to enhance readability and coherence. Several sentences are convoluted and require clarification.
The manuscript lacks sufficient citations to support the claims made by the authors. It is essential to include relevant references from peer-reviewed literature to back up the statements and hypothesis presented. The current reference list is limited and should be expanded to encompass the relevant field.
The authors have presented their hypothesis without considering alternative perspectives or counterarguments. It is crucial to acknowledge and address alternative explanations or theories related to the evolution of life and the genetic code. This will strengthen the manuscript's scientific rigor and ensure a balanced presentation.
The conclusion section is currently inadequate and does not provide a comprehensive summary of the findings or a clear statement regarding the significance of the proposed hypothesis. The authors should revise and expand this section to highlight the implications of their work and propose future directions for research.
Minor editing of English language required
Author Response
The reviewers recommend revisions and further clarification in the following areas:
- The introduction lacks a clear and concise background on the current understanding of the genetic code and its role in the evolution of life.
Our addition in Introduction:
As pointed out by Gross , Einstein՚s great advance was to put the symmetries as a dominant concept in the fundamental laws of physics, to regard the symmetry principle as the primary feature of nature: the symmetry principles dictate the form of the laws of nature. The Einstein՚s paradigm implies, in general, a broader view on the problem of evolution of natural laws. For example, the law of energy conservation is a natural consequence of existence of time symmetry, and not of some kind of evolution. Analogically, according to Einstein՚s paradigm, the SSyGC, which is unchangeable during whole evolution could be considered as a natural consequence of physicochemical symmetries with the corresponding mirror symmetry, and not being generated by an evolutionary process.
In 1943 Schrӧdinger proposed in his lecture at the Trinity College in Dublin that the hereditary material must take the form of an “aperiodic crystal”, implying the presence of symmetries in the structure of DNA. Ever since the Nirenberg՚s discovery in 1961 which codons code for individual amino acids, scientists searched for symmetries within genetic code. Up to the present discovery of the SSyGC, complete symmetry in the genetic code has not been found, leaving a doubt as to whether the symmetrical nature as the “protector” of order even exists.
In the framework of symmetry as one of the guiding principles, we discuss symmetry investigation of DNA molecule and of genetic code based on programmable biomolecular mediated processes and physicochemical laws, with the Einstein՚s symmetry paradigm extended to the life science. More specifically, here we discuss how can the laws of DNA and genetic code be considered as being related to the underlying physicochemical and mirror symmetries.
- It is essential to provide a comprehensive overview of the existing literature to contextualize the significance of the proposed DNA quadruplet 2 symmetry and the SSyGC. Please provide a more detailed literature review in this section. While the hypothesis is intriguing, the manuscript lacks a strong theoretical framework to support DNA 2 quadruplet symmetry and the SSyGC as the factor in the evolution of life.
Our answer:
The overview of included literature refers to the “strand symmetry” of DNA (Chargaffʼs second parity rule) since 1968 as an accepted and well-known model [1], and DNA quadruplet symmetry was introduced in our publications [9, 10, 26]. In the next sections, we compare here the views on symmetries of DNA molecule and genetic code symmetries with our discoveries during the last ten years. We postulate “the symmetry theory of the genetic code” which is based on the unique physicochemical purine – pyrimidine symmetry net between bases and codons of our SSyGC table and unchangeable during evolution, what is opposite of the concept of genetic code evolution through “progressive symmetry breaking theory” [46-53, 85]. At the end of this review, we propose in the fields of genetics, biology, chemistry, physics, medicine, and philosophy to investigate the origin of life in this framework.
Our additional text in the section “The Supersymmetry genetic code table”:
The intriguing algebraic approaches to the genetic code evolution through progressive symmetry breaking theory explained the observed degeneracy of the genetic code using the mathematical technique for organizing the group-theoretical structure [46-53, 85].
In conclusion, such degeneracy distribution through progressive symmetry breaking theory takes into consideration only one input, the number of codons for each amino acid according to the Nirenberg’s empirical result for degeneracies, but without assigning individual codons to amino acids and without finding physicochemical symmetries of the genetic code.
Evolution of the genetic code and life on Earth was a scientific challenge for many scientists, with interesting results, but without discovering complete physicochemical genetic code symmetries [2, 31, 34, 43, 45, 47, 48, 62, 63, 88, 92, 104, 105].
Our addition in Discussion:
The recent discovery of extraterrestrial abiotic amino acids with identical physicochemical symmetries of the genetic code and DNA molecule, and at the same time discovery of unchangeable genetic code during evolution for all RNA and DNA species is a challenge for scientists of genetics, biology, chemistry, physics, medicine, and philosophy to investigate and reveal the riddle of the origin of life.
- Additionally, including experimental or computational evidence would significantly strengthen the arguments put forth. The manuscript does not present any experimental data or simulations to support the proposed hypothesis. It is crucial to provide experimental validation for computational modeling results to demonstrate the feasibility and functional significance of the DNA quadruplet symmetry and the SSyGC. Without such evidence, the hypothesis remains speculative.
Our answer:
We presented a complex computational modeling to demonstrate the feasibility and functional significance of the DNA quadruplet symmetry in our article: Rosandić, M., Vlahović, I., Pilaš, I., Glunčić, M., Paar, V. An Explanation of Exceptions from Chargaff’s Second Parity Rule/Strand Symmetry of DNA Molecules. Genes 2022, 1950223, doi.org/10.3390/genes13111929 [10]. To illustrate statistically our results, an extensive Supplementary Materials of 350 pages was supplemented to that article.
We are using the same code for mapping 64 codons to 20 natural amino acids plus stop signals as determined experimentally by Nirenberg et al. [30], but instead of presenting this assignment in the Standard Genetic Code (SGC) table, we present the standard code in our Supersymmetry Genetic Code (SSyGC) table. The SGC table was formed as the 4×4 box pattern, with identical UCAG arrangement for the third base in each codon. With such construction, the total physicochemical symmetry of Nirenberg՚s code remained hidden. On the other hand, the SSyGC table formed from the 8×2 box pattern, with the third base in neighbouring pair of codon boxes being in Watson-Crick direct-complement (codon – anticodon) relation. With such construction the double mirror symmetry is also presents and, the total physicochemical symmetry of Nirenberg՚s code is automatically revealed. The symmetry structure of the SSyGC table enables an automatic transformation with directly alignment of all codons from direct boxes (top strand) and all codons from complement boxes (bottom strand) of the genetic code into a DNA-type. The SSyGC table is common for more than thirty different genetic codes (an example in Figure 6) and is identical with its purine-pyrimidine symmetry net for all RNA and DNA living species. The SSyGC table consists of all natural elements of code (64 codons for 20 natural amino acids and 3 stop signals). Because of that, it is not crucial to provide experimental validation for computational modelling to demonstrate feasibility and functional significance of the SSyGC table.
Our addition in the chapter: DNA quadruplet symmetry:
We show that the logarithmic relationship between oligonucleotide order and minimal DNA sequence length about 100 000 bp to establish the validity of CSPR automatically follows from the quadruplet structure of genomics sequence (Figure 3) [10]. Performing our quadruplet frequency analysis of all complete human chromosomes, for a random 200 000 bp sequence of each chromosome, and for the Neuroblastoma Break Point Family (NBPF) genes which code for Olduvai protein domains in the human genome [107], we show that the coding part of DNA (less than 2% of the whole genome, ~ 17 000 000 bp in chromosome 1) violates the CSPR. Opposite of that, the 98 % non-coding part and the whole human genome agree with CSPR as well as with DNA quadruplet symmetry.
- The overall structure and clarity of the manuscript need improvement. The authors should reorganize the content into distinct sections including an Introduction, Materials and Methods, Results, Discussion and Conclusion. Additionally, the writing style should be refined to enhance readability and coherence. Several sentences are convoluted and require clarification.
Our answer:
The manuscript is written in the form of review, where the sections like Materials and Method, and Results are not necessarily required. Following recommendation by the reviewer, several sentences are modified to improve the clarity.
Corrected sentences (red) in the section: The Supersymmetry Genetic code table
The genetic code is degenerate because more than one type of codon (2, 3, 4 or 6) may encode a single amino acid. The intriguing algebraic approaches to the genetic code evolution through progressive symmetry breaking theory explained the observed degeneracy of the genetic code with a mathematical technique for organizing the group-theoretical structure [46-53, 85]. Evolution of the genetic code through progressive symmetry breaking proposes that in the beginning it was not possible to distinguish the function of codons, which therefore all encode the same information. With consecutive creation of amino acids during such proposed evolution, the symmetries among codons, i.e., the number of codons within degeneracy groups gradually decrease (2 singlets, 9 doublets, 2 triplets, 5 quadruplets and 3 sextets). Because of that the symmetric pattern of codon ldegeneracy is supported with the unified mathematical framework by using the group-theoretical structure [54, 55]. In conclusion, such degeneracy distribution through progressive symmetry breaking theory takes into consideration only one input -– the number of codons for each amino acid - according to Nirenberg’s empirical result, and without any physicochemical affinity between codons and amino acids do not show any symmetry.
Our SSyGC table consists also of 16 codon boxes, as SGC, but in the form of 2 columns, each with 8 boxes, and with the same codons in each box like the SGC. The main difference in the SSyGC table is the position of the third base between direct and complement codon boxes on the principle of Watson-Crick pairing: GACU ↔ CUGA, AGUC ↔ UCAG what leads to the physicochemical code symmetry with the double mirror symmetry. (Figure 5).
For example, all symmetric codons such as UUU - AAA, CCC - GGG, or CAC - GUG, UGU – ACA etc. aligned strictly one below the other in all four rows because of mirror symmetry.
- The manuscript lacks sufficient citations to support the claims made by the authors. It is essential to include relevant references from peer reviewed literature to back up the statements and hypothesis presented. The current reference list is limited and should be expanded to encompass the relevant field.
Our answer:
According to the instruction by the Assistant Editor, for the review form of manuscript, at least 51% recent references should have been published during the last five years. Our choice of references [65 - 107] is related to investigations based on the Standard Genetic Code. References [1 - 64] are strong back up to statements and hypothesis presented. Four additional references are added. Additional references:
- Ikehara, K. Towards revealing the origin of life – Presenting the GADV hypothesis. Springer Nature, Cham, Switzerland 2021.
- Ikehara, K. Towards revealing the origin of life. Genes 2023, doi.org/10.3390/frai.2023.1128153.
- Yan Liu, Xuezhao Zhong, Zi Luo, Xsiangqui Meng, Rui Li, Wa Zhong, Lin Yang, Hualei Wang, Dongzhi Wei. The identification of a robust leucine dehydrogenase from a directed soil metagenome for efficient synthesis of L-2-aminobutyric acid. Biotechnology Journal 2023, doi.org/10.1002/biot.202200590.
- Glunčić, M., Vlahović, I., Rosandić, M., Paar, V. Tandemly repeated NBPF HOR copies (Olduvai triplets): Possible impact on human brain evolution. Life Science Alliance 2022. Doi.org/10.26508/lsa.202101306.
- The authors have presented their hypothesis without considering alternative perspectives or counter arguments. It is crucial to acknowledge and address alternative explanations for theories related to the evolution of life and genetic code. This will strengthen the manuscripts scientific rigor and ensure a balanced presentation. The concluding section is currently inadequate and does not provide a comprehensive summary statement regarding the significance of the proposed hypothesis. The authors revise and expands this section to highlight the implications of their work and propose future directions for research.
Our answer:
To better presented our “symmetry theory of genetic code” we added a new chapter:
What has during sixty years hindered the discovery of complete physicochemical symmetries of the genetic code?
In 1961 Nirenberg and collaborators deciphered the genetic code, determining experimentally which codons correspond to each of 20 natural amino acids, but without considering codon’s regularity in the form of genetic code table. That challenge was addressed by Nobel laureate F. H. Crick, and in 1968 a solution was published [31] under the name Universal Genetic Code table and readily included in biology and genetics textbooks. In search for symmetry in that genetic code table, the guideline was Watson-Crick A↔T and C↔G base pairing, that was discovered in 1953 for the structure of DNA molecule. This goal was achieved as a half-way result only, and the whole concept of creation of the genetic code table was comparison completed with the random Crick’s “frozen accident hypothesis”.
In the meantime, more than thirty different codes have been discovered for genomes of some bacteria and archaea as well as for some organellar mitochondrial and eukaryotic nuclear genomes. Because of that the name of the Universal Genetic Code was changed to the Standard Genetic Code.
The fact that the Standard Genetic Code is degenerate, i.e., that more than one codon can code for the same amino acid, the search for symmetries cannot be completely successful by amino acids arrangement. We realized that the key to the genetic code symmetries must be searched between codon purines and pyrimidines as the starting point. This approach led us to discovery of the Supersymmetry Genetic Code (SSyGC) table, characterized by codon physicochemical symmetries, also including the double mirror symmetry. The symmetry core of the SSyGC table is purine – pyrimidine symmetry net as a “golden rule”, which is common for all RNA and DNA species and unchangeable during evolution.
The Standard Genetic Code has ordering of codons according to pyrimidines (U, C) and purines (A, G). This ordering is strictly established both in horizontal and vertical directions. Thus, the resulting genetic code has only alphabetic symmetry, with UCAG ordering of bases. Therefore, the third base in codons does not differentiate among sixteen codon boxes in the Standard Genetic Code table and cannot fully contribute to discovery of physicochemical symmetries of genetic code. That has to this day contributed to the problem of many studies based on Standard Genetic Code table with UCAG ordering of the third base [65-102]. For example, the hydrophobicity and lipophilicity of amino acids have some symmetry relationship only with second base of codons [43]. The result was also studied by algebraic approaches to the degeneracy of genetic code and hypothesis of evolution of the genetic code through progressive symmetry breaking [46-54].
On the contrary, with our SSyGC table we have proved that the third base is crucial point for discovery of physicochemical symmetries on the principle of Watson-Crick base pairing (A↔U and C↔G like codon/anticodon) and for discovery of double mirror symmetry.
We have also proved that the DNA quadruplets have the same symmetries, as well as our classification of trinucleotides/codons (Figure 1). It is very important and unique for our SSyGC table that the symmetries of the SSyGC table are organized on the principle of direct-complement (codon-anticodon) what enables the direct transformation of the SSyGC table into the DNA molecule with also doble-mirror symmetry.
Because of complete physicochemical symmetries of the SSyGC table it is not necessary to involve “the frozen accident”. All more than thirty alternative genetic codes with slight departure from the standard code, can be incorporated in the SSyGC table (Figure 6). Thus, the SSyGC table fulfills all physicochemical criteria on the origin of the genetic code (2, 34, 62). Of special importance is the meaning of complete physicochemical symmetries in the SSyGC table, common for all living RNA and DNA species and unchanged during the whole evolution. In this theoretical approach, there is no evolution of the genetic code, but instead it has a power of natural law in analogy to Nӧther՚s theorem [5] for the natural law of energy conservation.
Our symmetry-based theory of genetic code broadens horizon for understanding evolution as a
fundamental process in creation and richness of life on Earth. This approach points out that besides mutation and natural selection also other factors may have been responsible for evolution, as are basic and unchangeable physicochemical purine-pyrimidine symmetry net as “the golden rule” of the genetic code structure.
We point out that the fundamental role of symmetry in the genetic code is to decrease disorder
(Information entropy) and to preserve integrity of biological system [8 -10, 26, 32, 33] during evolution in a way of extending Einstein՚s paradigm to put symmetries as a dominant concept in the fundamental law of physics to the phenomenon of life. One impressive case of realization of Einstein’s paradigm in physics is the famous Nӧther theorems: Emy Nӧther proved mathematically that the law of energy conservation is a consequence of the time symmetry [5].
Reviewer 2 Report
The authors have created new genetic code table based on symmetry between 64 triplet codons and have provided an interesting Super Symmetry Genetic Code (SSyGC) table in the article. It is surely important to create such a new genetic code table. However, the new genetic code table should not be of great significance, if the origins and evolutionary processes of the genetic code and protein could not be newly found from the new genetic code table. Therefore, I would like to recommend to the authors to revise the manuscript according to the major and minor comments, if the comments are valid.
Major comments
1. Why must the SSyGC table start with AUG start signal? The authors should show the reason why the genetic code originated from AUG start codon, if the authors consider so?
2. Can fundamental properties about the genetic code be found from the new genetic code tables, for example the origin and the genetic code, the order of codon capture and so on.
3. It is described in Discussion of the article that all 20 natural amino acids were delivered to the early Earth at the time of origin of life. Then, the authors should explain the reasons why non-natural amino acids, such as L-a-2-aminobutylic acid, which should be also delivered from space, are not used in the genetic code or in natural proteins? Can the authors explain the reasons?
4. It is considered that the first immature genetic information (genetic code) for synthesis of an immature [GADV]-protein was written in RNA and DNA with anticodons GNCs carried by anticodon stem-loop tRNAs (see the references, for example. 1. K. Ikehara; Towards Revealing the Origin of life.—Presenting the GADV Hypothesis; Springer Nature, Gewerbestrasse: Cham Switzerland, 2021. 2. K. Ikehara; Genes 2023, 14, 375. https://doi.org/ 10.3390/genes14020375). Therefore, it seems to me that the SSyGC could be easily prepared by arranging triplet codons symmetrically in a new genetic code table, because the genetic code is composed of four bases, with which codon sequences are written in anti-parallel double-stranded (ds)-RNA (later ds-DNA) as genetic information carrier, which is composed of two base pairs (AT(U) and GC). How do the authors consider about the comment?
Minor comments
1. Line 384: Phrase “from an NA to DNA molecule” should be “from an RNA to DNA molecule”.
The English used is correct and readable,
Author Response
Major comments:
- Why must the SSyGC table start with AUG start signal? The authors should show the reason why the genetic code originated from AUG start codon, if the authors consider so?
Our answer in section: The Supersymmetry genetic code table
It is fascinating that the SSyGC table starts with a very important AUG start signal. This is an automatic result of the SSyGC symmetries. Namely, the position of each codon is unique and strictly localized. At the same time all three stop signals (UAG, UAA, TAG) are positioned according to the mirror symmetry in both halves of the code in two boxes with the same 1-0-0 purine-pyrimidine relationship (Figure 5).
Namely, our path to ordering of codons started from serine, because we noticed the unique regularity: its 6 codons are in two neighboring boxes which are in direct-complement relation. In this way we discovered the logic of the third base in codons and the link between Serine and Arginine in the same column of boxes and between Serine and Leucine in the neighboring column of boxes. After identifying this core structure of the SSyGC table, the further ordering of other codons according to a similar logic is straightforward.
- Can fundamental properties about the genetic code be found from the new genetic code tables, for example, the origin and the genetic code, the order of codon capture and so on.
Our detailed answer is in the new secion: What has during sixty years hindered the discovery of complete physicochemical symmetries of the genetic code? (Page 4 and 5).
- It is described in Discussion of the article that all 20 amino acids were delivered to the early Earth at the time of origin of life. Then the authors should explain the reasons why non-natural amino acids, such as L-a-2 aminobutyric acid, which should be also delivered from space, are not used in the genetic code or in natural proteins? Can the authors explain the reasons?
Our answer in the Discusion:
Very important discovery on asteroid is also the presence of L-2-aminobutryc acid as a chiral precursor for the synthesis non-natural L-amino acids in the later stage of origin [104 - 106]. The discovery of extraterrestrial abiotic amino acids confirms our hypothesis that for the origin of life on Earth all 20 natural amino acids were necessary. There are scientific arguments that they were already present at the time of the creation of Solar system [61].
Our answers in the section: The Supersymmetry Genetic Code table:
Translation is not limited to twenty amino acids. The additional Selenocysteine as the 21st amino acid takes possession of the UGA stop signal and Pyrolysine as the 22nd amino acid [56] takes the UAG stop signal, but the SSyGC table and the basic purine – pyrimidine symmetry net remained unchanged.
One could assume that also some new non-natural amino acid, because of a special metabolic need of some species (or experimentally), adopts the codon of some other natural amino acid or stop signal of code without disturbing the physicochemical symmetry net of the SSyGC table. However, there is a question of the consequences of such artificial translation, with possible harmful or even lethal consequences.
- It is considered that the first immature genetic information (genetic code) for synthesis of an immature (GADV) – protein was written in RNA and DNA with anticodons GNCs carried by anticodon stem-loop tRNAs (See references 1. K. Ikehara; Towards revealing the origin of life – Presenting the GADV hypothesis, Springer Nature 2021, 2. K. Ikehara, Genes 2023). Therefore, it seems to me that the SSyGC could be easily be prepared by arranging triplet codons symmetrically in a new genetic code table, because the genetic code is composed of four bases, with which codon sequences are written in anti-parallel double stranded (ds)-RNA (later ds-DNA as genetic information carrier, which is composed of two base pairs (AT(U) and GC). How do the authors consider the comment?
It seems that the codon – anticodon symmetry as a complete structural physicochemical form of the unique SSyGC table made it possible in the early Earth synthesis of an immature proteins by anticodon stem-loop tRNA and later to double stranded DNA, what is also theoretical approach from earlier investigations [104, 105].
Minor comments:
Line 385: “Phrase” from NA to DNA is corrected to: from RNA to DNA
Round 2
Reviewer 2 Report
The authors have well revised the previous manuscript. However, I have found some minor points in the revised manuscript, which should be re-revised. So, I would like to recommend the editor to accept the re-revised manuscript after re-revisions according to my minor comments described below.
Minor comments
1. Line 65-66. Font size of the phrase “Einstein՚s symmetry paradigm extended to the life science” should be revised.
2. Line 348: “TAG” should be “UGA”.
3. Line 545: “L-2-aminobutryc” should be “L-2-aminobutyric”.
4. Line 856: “105. Ikehara, K. Towards revealing the origin of life. Genes 2023, doi.org/10.3390/frai.2023.1128153.” should be “105. Ikehara. K. Why were [GADV]-amino acids and GNC codons selected and how was GNC primeval genetic code established? Genes 2023, doi.org/10.3390/genes14020375”.
Some mistypes described in the comments and suggestions for authors should be revised.
Author Response
Dear Reviewer,
Thank you for your very thorough evaluation of the manuscript. All your comments have been included in the last corrected version.
Warm regards,
Marija Rosandić